



# Measurement report: Mass and Density of Individual Frozen Hydrometeors

Karlie N. Rees[1], Dhiraj Singh[2], Eric R. Pardyjak[2], and Timothy J. Garrett[1]

[1]Department of Atmospheric Sciences, Salt Lake City, UT, USA
[2]Department of Mechanical Engineering, Salt Lake City, UT, USA

**Correspondence:** Tim Garrett (tim.garrett@utah.edu)

**Abstract.** A new precipitation sensor, the Differential Emissivity Imaging Disdrometer (DEID), is used to provide the first continuous measurements of the mass, diameter, and density of individual hydrometeors. The DEID consists of an infrared camera pointed at a heated aluminum plate. It exploits the contrasting thermal emissivity of water and metal to determine individual particle mass by assuming that energy is conserved during the transfer of heat from the plate to the particle during

evaporation. Particle density is determined from a combination of particle mass and morphology. A Multi-Angle Snowflake Camera (MASC) was deployed alongside the DEID to provide refined imagery of particle size and shape. Broad consistency is found between derived mass-diameter and density-diameter relationships and those obtained in prior studies. However, DEID measurements show a generally weaker dependence with size for hydrometeor density and a stronger dependence for aggregate snowflake mass.

## 1   Introduction

Predictions of precipitation amount, location, and duration have been shown to be especially sensitive to parameterized expressions for how fast a hydrometeor falls (Rutledge and Hobbs, 1984; Reisner et al., 1998; Hong et al., 2004; Fovell and Su, 2007; Lin et al., 2010; Liu et al., 2011; Iguchi et al., 2012; Thériault et al., 2012), affecting forecasts of hurricane trajectories (Fovell

and Su, 2007) and storm lifetimes (Garvert et al., 2005; Colle et al., 2005; Milbrandt et al., 2010). From the perspective of fluid dynamics, fallspeed can be related to the mass and density of precipitation particles (Böhm, 1989). Observationally, one of the most frequently cited datasets that lies at the heart of current bulk microphysical parameterizations (e.g. Reisner et al., 1998; Hong et al., 2004; Tao et al., 2003) comprises just 376 snowflakes captured and photographed in the Cascade mountain range. Individual hydrometeors were melted on a sheet of plastic film, from which relationships were obtained between hydrometeor

mass, fallspeed, and diameter as a function of particle habit, and in the case of graupel, density (Locatelli and Hobbs, 1974). Despite its limited scope, later numerical studies (Böhm, 1989; Khvorostyanov and Curry, 2002; Heymsfield and Westbrook, 2010; Kubicek and Wang, 2012) and ground-based disdrometer measurements (Kruger and Krajewski, 2002; Barthazy et al.,





2004; Yuter et al., 2006; Newman et al., 2009) have lent general support to this reference data set, although concerns remain about geographic and temporal specificity, measurement limitations (Yuter et al., 2006; Battaglia et al., 2010), and the ability

to quantify the extent of riming (Barthazy and Schefold, 2006; Brandes et al., 2008).

Particle density measurements have proven more difficult to ascertain as they require, in addition to mass, an estimate of particle volume. This may be reasonably obtained for quasi-spherical particles such as lump graupel (Locatelli and Hobbs, 1974), but the task is considerably more challenging when snow particles are formed from ice crystal aggregation. A possible approach is to infer density from fallen snow using column measurements (Conger and McClung, 2009), capacitance probes

(Dent et al., 1998), or a combination of a camera for snow depth and an electric scale for snow mass (Muramoto et al., 1995). Muramoto et al. (1995) and Brandes et al. (2007) measured the bulk mass of snowflakes using a weighing gauge, from which the bulk volume was determined with 2D camera imagery of individual snowflakes. Tiira et al. (2016) determined a volume flux weighted snow density for a population of snowflakes using particle size distribution, fallspeed, and a weighing gauge to estimate the mass, and 2D camera imagery to determine effective diameter and volume. However, snow undergoes compaction

and melting on the ground, so the relationship to individual particle density in the air is approximate (Brun et al., 1992).To obtain individual snowflake particle density, Magono and Nakamura (1965) collected individual wet and dry snowflakes on a piece of dyed filter paper, from which the outline of the flake was manually measured. Individual volume was inferred from the major and minor axes of the outline of snow, and the mass from the outline of the melted snowflake. Holroyd (1971) also made measurements of the major and minor axes of powder snow and dendrites from (Magono and Nakamura, 1965). A limitation

of these datasets is that they were necessarily small given the manual nature of the effort.

Here, we present continuous measurements of individual the masses of frozen hydrometeors using a new instrument, the Differential Emissivity Imaging Disdrometer (DEID). DEID data are combined with photographic imagery obtained using a Multi Angle Snowflake Camera (MASC) (Garrett et al., 2012) to obtain estimates of particle density.

## 2   Methods

All measurements described in this study were acquired at a meteorological measurement tower placed at the mouth of Red Butte Canyon (40.76857, -111.82614) in Salt Lake City at 1547 m elevation in January and February 2020. More details about the site and measurement campaign are provided by Singh et al. (2021).

### 2.1   Particle mass from thermal imaging

The Differential Emissivity Imaging Disdrometer (DEID) (Singh et al., 2021) consists of a thermal camera operating at a

frequency between 2 and 12 Hz pointed at an aluminum plate placed atop a hotplate maintained at a self-sustained temperature of 85°C. The camera distinguishes hydrometeors as they melt and evaporate as white regions on a black background (Figure 1) due to the contrasting infrared emissivities of water ($\epsilon \approx 0.96$) and aluminum ($\epsilon \approx 0.03$). A small strip of polyimide tape ($\epsilon \approx 0.95$) is applied to the aluminum plate to provide a reference temperature, as well as a pixel to length dimension conversion based on the tape's known width. The effective collection area of the DEID was $A \approx 7\,\text{cm} \times 5\,\text{cm}$ and the per pixel resolution





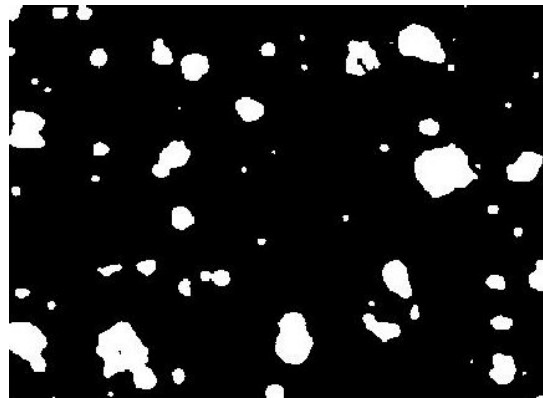

**Figure 1.** Thermal camera imagery from the DEID, showing regions of high emissivity water (white) on a low emissivity aluminum plate (black).

of imaged particles $p$ was 190 $\mu$m. Processing of the thermal camera imagery yields the hydrometeor area on the plate, the temperature difference between the water and the plate, and the evaporation time.

Individual particle mass is determined using the DEID by employing the assumption that the heat gained by a hydrometeor is equivalent to the heat lost by the plate during the process of melting and evaporation. The heat balance equation is:

$$c_p \Delta T \int dM + L_{eqv} \int dM = \int_0^t \frac{K}{H} A(t)(T_p - T_w(t)) dt \tag{1}$$

where $c_p$ is the specific heat capacity of water at constant pressure, $\Delta T$ is the difference in temperature between 0 and time, $t$, $M$ is the mass of the hydrometeor, and $L_{eqv}$ is the equivalent latent heat required for the conversion of the hydrometeor to gas. For liquid precipitation $L_{eqv} = L_v$ where $L_v$ is the latent heat of vaporization of water. For solid precipitation, $L_{eqv} = L_v + L_f$ where $L_f$ is the latent heat fusion for water. $K$ is the thermal conductivity of the aluminum plate, $H$ is the plate thickness, $A(t)$ is the cross-sectional area of the water droplet at time $t$, $T_p$ is the temperature of the hotplate, and $T_w(t)$ is the temperature of
the water at time $t$. Taking

$$\int dM = K_d \int_0^t A(t)(T_p - T_w(t)) dt$$

a single calibrated constant $K_d$ for the plate was determined experimentally by applying known masses of water from a micropipette to the plate. (See Appendix Afor the derivation of $K_d$). The heat balance equation was found in a laboratory setting to be highly insensitive to ambient winds, temperature, and humidity (Singh et al., 2021).

The temperature difference between the plate and water on the plate ($\Delta T = T_p - T_w(t)$) can be determined using the mean pixel intensity of the particle in the thermal camera imagery (see Appendix B). The mass calculation then simplifies to:

$$M = K_d \Delta T \int A(t) dt \tag{2}$$

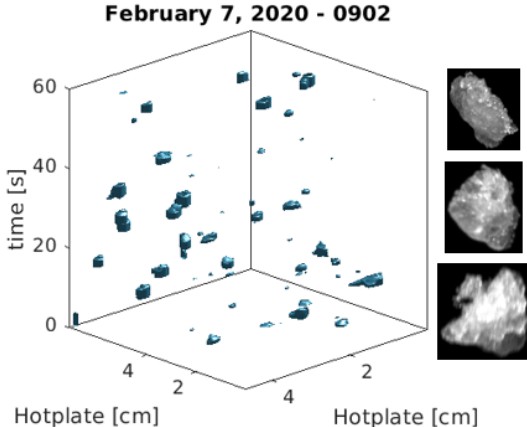

**Figure 2.** Evaporation volumetric profiles on the DEID plate in a space of imaged area and time $V_t = \int A(t)dt$ for graupel.

Three-dimensional regions integrated over particle area and evaporation time were constructed for each particle during the melting process defined by $V_t = \int A(t)dt$ so that from Eq: 2 $M = K_d V_t \Delta T$. An example of a one-minute time interval with 3D volumetric regions representing each individual particle is shown in Figure 2 for graupel. Additional 3D representations for other snowflake types are shown in Appendix B.

## 2.2 Hydrometeor photography

The Multi-Angle Snowflake Camera (MASC) (Garrett et al., 2012) houses three high-speed visible spectrum cameras arranged concentrically with a separation angle of $36°$. As a particle falls through the instrument collection aperture, two vertically spaced infrared detectors simultaneously trigger the cameras to take three simultaneous photographs of the particle from the side and to measure the fallspeed between the two detectors. The MASC software processes the imagery to calculate the surface area, geometric cross-section, perimeter, orientation, aspect ratio, complexity, flatness, and whether the particle is a raindrop (Shkurko et al., 2016). Images taken by the Multi-Angle Snowflake Camera (MASC) are used in conjunction with the DEID to confirm precipitation type and refine density measurements. A mosaic of MASC images from February 5 and 6, 2020 is shown in Figure 3. Additional snowflake imagery catalogued from each storm is shown in Appendix C.

## 2.3 Particle volume and density

Immediately following the arrival of a hydrometeor on the plate, the particle cross-sectional area increases rapidly as the hydrometeor adjusts from the ambient air temperature to the temperature of the plate, melting in the process. It reaches a maximum size $A_{max}$ before shrinking during evaporation. Although the particle may be melted at the moment that it reaches $A_{max}$, frozen particles nonetheless tend to maintain their shape such that $A_{max}$ is approximately representative of the frozen cross-sectional area in air. To obtain an estimate of particle density from particle mass, a spherical volume $V_S$ for each hydrometeor can be estimated from an effective diameter $D_{eff}$ derived from the maximum cross-sectional area $A_{max}$ $V_S = \frac{\pi}{6} D_{eff}^3$, where



**February 5-6, 2020**

**Figure 3.** A selection of MASC snowflake images obtained between February 5 and February 6, 2020.



$D_{\text{eff}} = \sqrt{\frac{4}{\pi} A_{\text{max}}}$. This geometric definition is consistent with that taken by Locatelli and Hobbs (1974), who prescribed $D_{\text{eff}}$ as "the diameter of the smallest circle into which the aggregate as photographed will fit without changing its density." Unless

otherwise specified, all instances of "diameter" in the text refer to $D_{\text{eff}}$. An alternative diameter metric is $D_{\text{max}}$ as defined by the circumscribed diameter from the maximum horizontal dimension of the hydrometeor as it lies on the hotplate. The measured relationship between these two metrics is described below.

For a more precise estimate of particle volume, side-viewing MASC imagery of hydrometeors can be used to determine the average aspect ratios of hydrometeors measured over a one-minute time interval. In general, by approximating hydrometeors as

an ellipsoid, the volume of a snowflake is $V_M = \frac{\pi}{6} D_{\text{max}} D_{\text{min}} D_v$ where $D_{\text{max}}$ is the longest dimension as seen by the DEID, $D_{\text{min}}$ is the shortest, and $D_v$ is bounded by the two. For example, if the hydrometeor shape is an oblate spheroid, then $D_v = D_{\text{max}}$, and if a prolate spheroid then $D_v = D_{\text{min}}$. As a snowflake falls onto the plate, the maximum dimension is expected to lie flat with respect to the plate. $D_{\text{eff}}$ is expected to lie between the maximum and minimum dimensions of the snowflake, so for the more general case of an ellipsoid, a reasonable assumption is that $D_v \simeq D_{\text{eff}}$.

The issue here is that the DEID is unlikely to provide a measure of the minimum dimension. However, it can be reasonably inferred from side views of hydrometeors provided by the multiple concentric images captured by the MASC. For a given snowflake, the minimum of the aspect ratios for each hydrometeor seen by the multiple MASC cameras as captured within a one minute interval and subsequently averaged $\alpha_{\text{min}}$, can reveal a characteristic minimum dimension in the vertical direction for the time period that is otherwise invisible to the DEID. Taking $D_{\text{min}} = \alpha_{\text{min}} D_{\text{max}}$, the MASC-adjusted volume is then

$$V_M = \frac{\pi}{6} D_{\text{max}}^2 D_{\text{eff}} \alpha_{\text{min}} \tag{3}$$

The spherical and mass-adjusted density of individual hydrometeors can then be calculated from the mass $M$ and volume as $\rho_S = M/V_S$ and $\rho_M = M/V_M$ respectively.

## 3   Results

Over the course of five storms that took place in Salt Lake City, Utah on January 14, 2020, January 17, 2020, January 26,

2020, February 2-3, 2020, and February 5-6, 2020, the DEID detected 132,459 individual hydrometeors. Of those hydrometeors, 104,812 were snowflakes, and the remainder were either rain or a rain-snow mix. A filtering algorithm rejected small hydrometeors with fewer than three contiguous pixels of data in all three dimensions (See Appendix B), leaving a total dataset of 109,316 hydrometeors, of which 86,285 were snowflakes and the remainder rain. Of those snowflakes, density estimates for 43,649 were obtained using corresponding MASC imagery. The density of rain of course is known to be $1000\ \text{kg m}^{-3}$

and would provide a valuable reference point for this study. Unfortunately, its measurements cannot be addressed using the techniques described here as raindrops do not preserve their area after impaction on the plate.





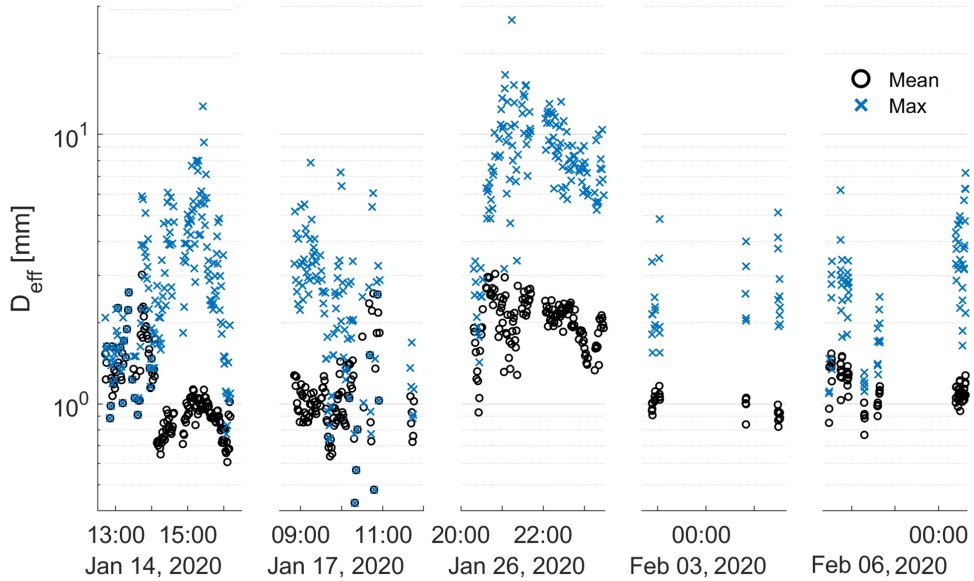

**Figure 4.** One-minute averaged maximum and mean effective diameters of measured hydrometeors

## 3.1 Time series

Figure 4 shows the mean and the maximum effective diameters of the hydrometeors considered for each one-minute sampling interval. The largest observed maximum diameters occurred during January 14 and January 26, when large aggregates were the primary snow type (See Appendix C for corresponding snowflake imagery). During such periods characterized primarily by aggregate snowflakes, a larger difference was observed between the mean and maximum diameter observed. Periods characterized primarily by graupel, including early January 14 and late January 17, exhibited smaller differences.

Adopting either the spherical volume or the MASC-adjusted volume, the probability density functions for $\rho_S$ and $\rho_M$ separated by storm and combined for all storms are shown in Figure 5. Derived density values are approximately log-normally distributed largely ranging between 10 and 100 kg m$^{-3}$. The calculated densities differ by approximately a factor of two depending on the calculation method used. For all storms, the mean spherical density was 38 kg m$^{-3}$ and the logarithmically-weighted mean was 35 kg m$^{-3}$, while the respective MASC-adjusted density values were 90 kg m$^{-3}$ and 70 kg m$^{-3}$. The relative absence in the distribution of high density values derived using the spherical approximation suggests an underestimate given that Locatelli and Hobbs (1974) found lump graupel to have densities ranging as high as 450 kg m$^{-3}$ and dense graupel has also been described by others (Magono and Nakamura, 1965; Holroyd, 1971; Muramoto et al., 1995; Fabry and Szyrmer, 1999; Brandes et al., 2007; Tiira et al., 2016). For example, for the period January 14 1243-1259 MST when MASC imagery showed graupel predominated during a period with temperatures near the melting point, the spherical calculation yielded an average density of 88 kg m$^{-3}$ versus 131 kg m$^{-3}$ for the MASC related estimates. The respective logarithmically-weighted mean values were 87 and 121 kg m$^{-3}$.





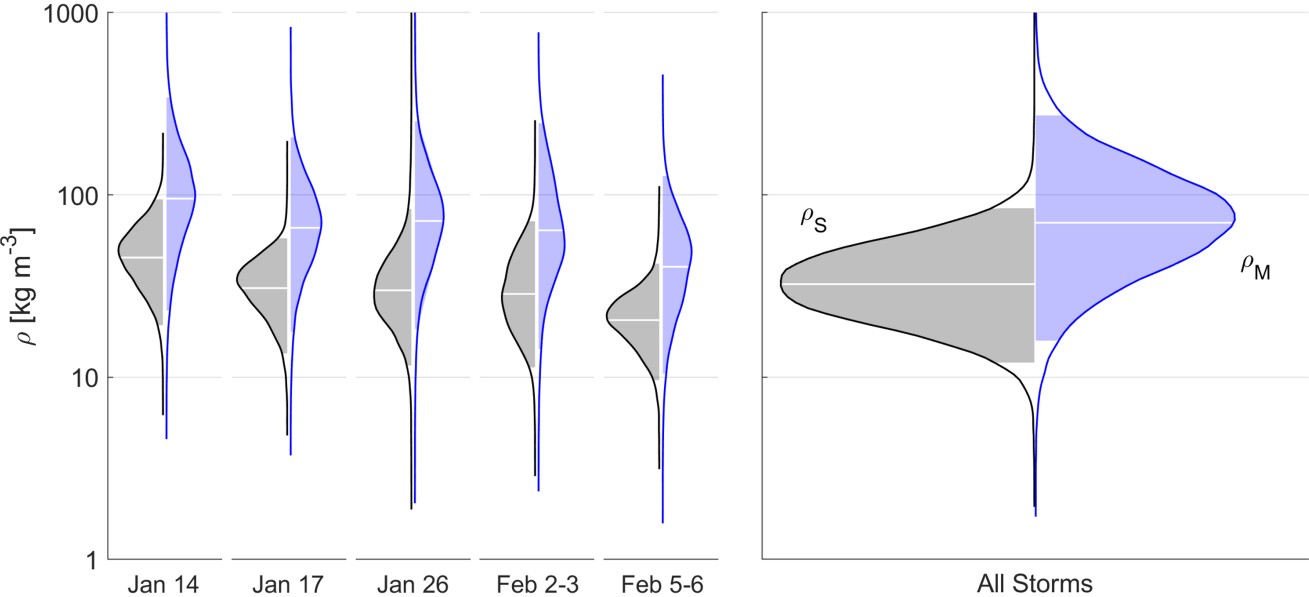

**Figure 5.** Linear probability density functions of snow density measurements using the spherical volume assumption (gray) and the MASC-adjusted volume (blue) separated into individual storms (left) and for all snowflakes (right). Mean values and $\pm 2\sigma$ values are represented by white horizontal lines and the boundary between shading and white space under the tails of the curves.

**Table 1.** Relationships between hydrometeor circumscribed and effective diameter.

| Type | $D_{\mathrm{max}} = aD_{\mathrm{eff}}^{b}$ | | $R^2$ |
|---|---|---|---|
| | a | b | |
| All | 1.17 | 1.11 | 0.93 |
| Graupel | 1.05 | 1.02 | 0.95 |
| Densely Rimed | 1.23 | 1.17 | 0.91 |
| Aggregates | 1.08 | 1.14 | 0.94 |

## 3.2 Diameter and aspect ratio

In the following sections, mass-diameter and density-diameter relationships are expressed with respect to $D_{\mathrm{eff}}$. By way of reference, for the 86,285 snowflakes included in this study, $D_{\mathrm{max}}$ can be correlated with $D_{\mathrm{eff}}$ through the power law relationship $D_{\mathrm{max}} = aD_{\mathrm{eff}}^{b}$ with values for $a$ and $b$ summarized in Table 1. In general, the two quantities are highly correlated, and the relationship is nearly linear. For the ensemble taken as a whole, the relationship is $D_{\mathrm{max}} = 1.17D_{\mathrm{eff}}^{1.11}$ with a square correlation coefficient of $R^2 = 0.93$. The average measured aspect ratios seen by the MASC for aggregates, densely rimed particles, and graupel were 0.62, 0.65, and 0.82 respectively.

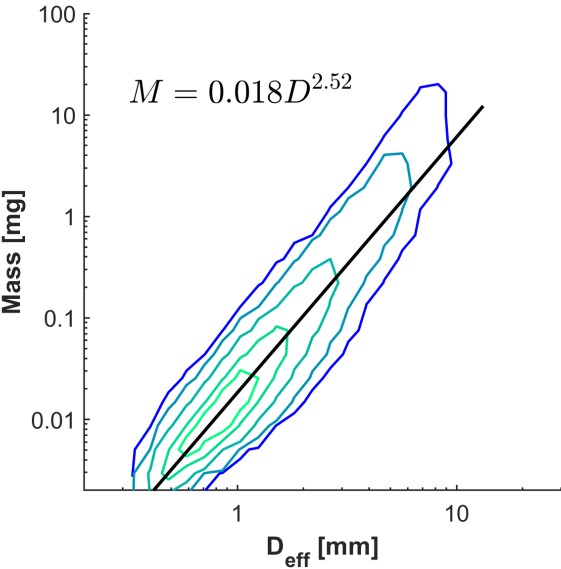

**Figure 6.** Mass-diameter relationship for all 86,285 snowflakes. Contours shown are for the 25th, 50th, 75th, 90th, and 95th percentiles.

### 3.3 Mass-Diameter relationships

The mass-diameter relationship for all snowflakes is shown in Figure 6. The prefactor observed in the mass-diameter relationship is 0.018, which is approximately consistent with the range of values described by Locatelli and Hobbs (1974) for the
densely rimed snowflakes (Table 2), comprising 47% of the snowflakes observed in our study. The exponent 2.52, however, generally exceeds those values obtained by Locatelli and Hobbs (1974) for graupel-like (2.1 to 2.4), densely rimed (2.1 to 2.3), and aggregated (1.4 to 1.9) snow.

Snowflakes were selected manually through visual classification using MASC imagery for time periods at least three minutes long where no other snow types were present, defined as aggregates (N=18,049), graupel (N=34), and densely rimed
(N=15,721). Graupel occurred frequently, although there was only one period 16 minutes long that did not exhibit any other type of snow. Mass-diameter relationships $M=aD^b_{eff}$ for graupel, densely rimed snow, and aggregates are compared with those found by (Locatelli and Hobbs, 1974) in Figure 7 and Table 2. Values of $a$ and $b$ for graupel determined by the DEID ($a$=0.047 and $b$=2.73) lie within the ranges observed by Locatelli and Hobbs (1974) $0.042 < a < 0.140$ and $2.6 < b < 3.0$. The quantity of densely rimed and aggregate snowflakes collected by the DEID was rouhgly five orders more numerous than those described
by Locatelli and Hobbs (1974). For densely rimed snowflakes, the prefactor $a$ lies within the range observed by Locatelli and Hobbs (1974) and the exponent $b$ is within 10%. However, aggregate snowflakes differ by 30% in the exponent and by 43% in the prefactor. Notably, the exponent value of $b = 2.75$ is markedly greater than obtained previously using various theoretical and observational approaches (Dunnavan et al., 2019).





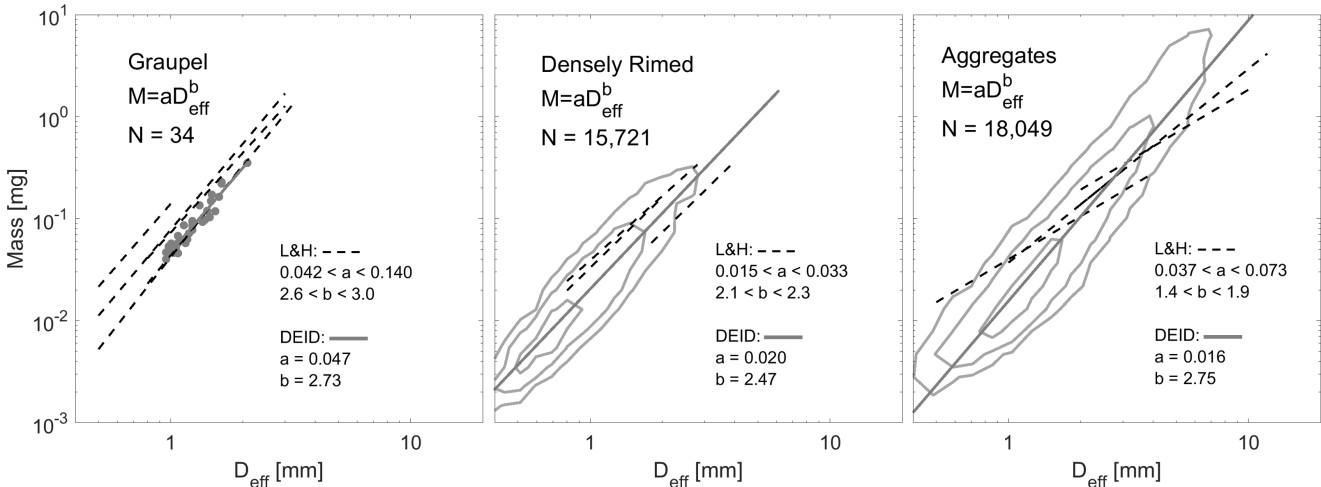

**Figure 7.** Mass-Diameter relationships from the DEID (solid lines) sorted by type according to MASC imagery and compared with fits from Locatelli and Hobbs (1974), dashed lines. Contours are for 25th, 50th, and 95th percentile bounds.

**Table 2.** Mass-Diameter relationship comparison

| Type | Locatelli & Hobbs (1974) | | | | DEID | | | |
|---|---|---|---|---|---|---|---|---|
| | $M=aD^b$ | | $R^2$ | N | $M=aD^b$ | | $R^2$ | N |
| | a | b | | | a | b | | |
| Graupel | 0.042 - 0.140 | 2.6 - 3.0 | 0.91 - 0.98 | 17 - 58 | 0.047 | 2.73 | 0.92 | 34 |
| Graupel-like Snow | 0.021 - 0.059 | 2.1 - 2.4 | 0.72 - 0.91 | 17 - 31 | | | | |
| Densely Rimed | 0.015 - 0.033 | 2.1 - 2.3 | 0.78 - 0.92 | 9 - 13 | 0.020 | 2.47 | 0.88 | 15,721 |
| Aggregates | 0.037 - 0.073 | 1.4 - 1.9 | 0.78 - 0.91 | 19 - 27 | 0.016 | 2.75 | 0.90 | 18,049 |

Mass-Diameter relationship $M=aD^b$ where mass is in milligrams and diameter is in millimeters

## 3.4 Density-Diameter relationship

Density-diameter relationships for all snowflakes in this study are shown in Figure 8 and a comparison of density-diameter relationships from this work and prior studies is shown in Figure 9. Using the spherical volume approximation (Figure 8, left), the measured values for density are rarely greater than 100 kg m$^{-3}$, suggesting a possible underestimate. Using a similar assumption, Muramoto et al. (1995) observed similarly low values of density (Figure 9). The density-diameter relationships from other studies shown in Figure 9 include both dry and wet snowflakes and ice particles with densities extending into the

200-300 kg m$^{-3}$ range for the lowest diameters observed.

More refined density calculations supplemented by MASC data are shown in Figure 8, right, and include high values near the density of bulk water, as would be expected for wet snowflakes that have partially melted before reaching the hotplate. The





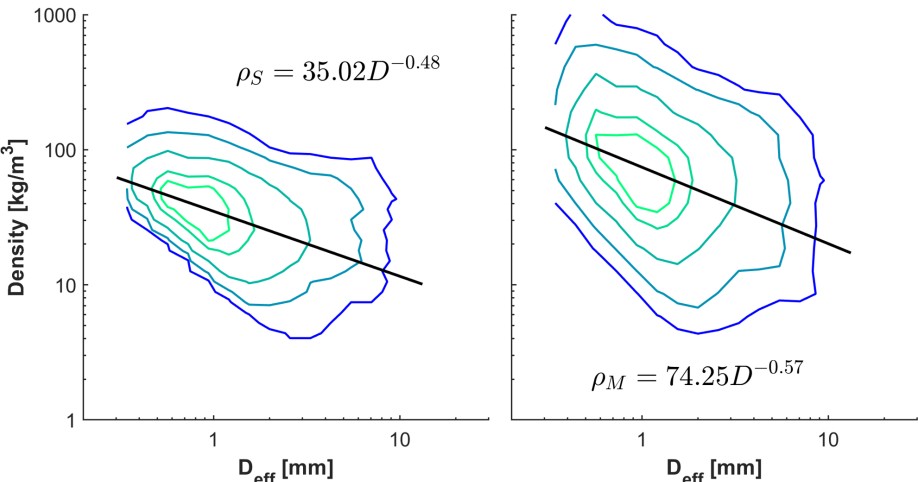

**Figure 8.** Density-diameter relationships for spherical (left) and MASC-adjusted (right) volume for all snowflakes. Contours shown are for the 25th, 50th, 75th, 90th, and 95th percentiles.

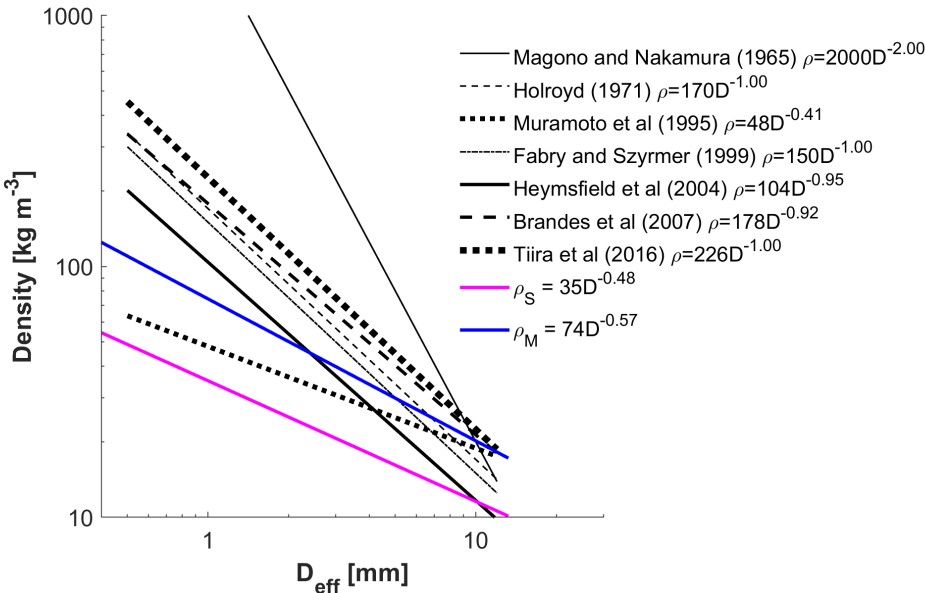

**Figure 9.** Density-diameter relationships from previous studies and those obtained here using the spherical density method (magenta), and MASC-adjusted density method (blue).

prefactors $a$ are 35.02 and 74.25 for the spherical and MASC-adjusted density respectively and the exponents $b$ are -0.48 and -0.57.





## 4  Discussion

A notable difference between the mass-diameter relationships from our study versus those described previously by Locatelli and Hobbs (1974) lies in fitted exponents for aggregate type snowflakes. In the earlier study these ranged from 1.4 to 1.9, while our results point to an exponent of 2.75, very similar to that observed for graupel and densely rimed snowflakes. An added implication of the larger exponent is that the masses of very small aggregates with diameters less than 1 mm are generally smaller than have previously been reported (Figure 7). While the MASC-adjusted density-diameter relationships align closely with several previous studies (Figure 9) for particle sizes larger than approximately 5 mm diameter, much lower values of density tend to be observed at smaller particle sizes. Where the exponents described in previous studies (Holroyd, 1971; Fabry and Szyrmer, 1999; Heymsfield, 2003; Brandes et al., 2007; Tiira et al., 2016) are close -1, those from our study are approximately -0.5, suggesting a weaker dependence of density on particle size than is generally assumed. Indeed, even for rimed snowflakes, high resolution electron microscope imagery reveals a highly porous internal structure (Rango et al., 2003; Enzmann et al., 2011). On the ground, the density of snowfall at a high-elevation location in the Wasatch Mountains in Utah is typically less than $100 \, \mathrm{kg \, m^{-3}}$ (Alcott and Steenburgh, 2010), a location where mode snowflake diameters lie between $1 \, \mathrm{mm}$ and $2 \, \mathrm{mm}$ (Garrett and Yuter, 2014).

Combining the weak dependence of density with size and the large exponent near 3 of the mass, a speculation may be made that frozen hydrometeors, while topologically complex, maintain a nearly Euclidean scaling with their maximum dimension as they grow. The elemental crystal or frozen droplet that comprises the hydrometeor structure is not obviously affected by the collection process. So, even if the hydrometeor ultimately attains a highly non-spherical structure as it falls, the internal porosity of the structure may be expected to remain approximately fixed.

## 5  Summary

We describe measurements of the mass and density of individual frozen hydrometeors obtained using a new instrument, the Differential Emissivity Imaging Disdrometer. Power-law mass-diameter relationships obtained by the DEID derived from 86,285 measured particles agree well with widely used relationships published by Locatelli and Hobbs (1974), which was based on a much more limited dataset. The exception is that aggregate-type flakes measured by the DEID have a significantly higher exponent close to a value of 3. To obtain hydrometeor density from the measured mass, estimates of volume are required. Here, a simple spherical approximation for particle volume based on the particle equivalent diameter seen by a thermal camera viewing the heated plate led to density estimates approximately a factor of two lower than those using a more refined calculation that incorporated concurrent MASC measurements of particle aspect ratio. For the subset of DEID measurements that included coincident MASC imagery totalling 43,649 hydrometeors, the resulting density-diameter relationships suggest substantially lower densities of particles smaller than 5 mm than has been observed in most prior studies. It may be that existing bulk microphysical parameterizations in numerical weather models tend to underestimate the masses of large frozen hydrometeors while overestimating those of smaller hydrometeors. If true, any revision could have possible implications for forecasts of





snow water deposition in mountain reservoirs. Future anticipated refinements to the DEID particle volume algorithm at a wider range of locations may help further refine estimates of hydrometeor density.

**Appendix A: Hotplate calibration**

The constant $K_d$ includes the specific heat capacity, equivalent latent heat, and the plate's thickness and thermal conductivity, was calibrated experimentally for the DEID aluminum plate (Singh et al., 2021). The plate was roughened with 600 grit (P1200) sandpaper to allow for droplet spreading and more rapid evaporation. Water drops of known masses were evaporated on the plate to determine the calibrated coefficient. Combining the constants in Equation (1) yields,

$$(c_p\Delta T_{\mathrm{ev}} + L_v)M = \frac{K}{H}\int_0^t A(t)(T_p - T_w(t))dt \tag{A1}$$

where $M$ is a known mass of water. A constant is determined that includes the thermal conductivity and the thickness of the hotplate using droplets of known mass $M = 2 \pm 0.2 \times 10^{-5}$ kg:

$$\frac{K}{H} = \frac{M(c_{p,w}\Delta T_{\mathrm{ev}} + L_v)}{\int_0^t A(t)T_p - T_w(t)dt} \tag{A2}$$

where $\int_0^t T_p - T_w(t)A(t)dt$ was determined from DEID measurements, $c_{p,w} = 4.28 \times 10^3$ J K$^{-1}$ kg$^{-1}$ is the specific heat of water at constant pressure, $\Delta T_{\mathrm{ev}} = 100$K, and $L_v = 2.26 \times 10^6$ J kg$^{-1}$ is the latent heat of vaporization. Determined through

10 trials, $K/H = 4.1 \times 10^3$ kg s$^{-3}$ K$^{-1}$.

Including the latent and specific heat required to evaporate liquid water and ice, respectively, the derived values of $K_d$ for liquid and ice are then:

$$K_{d,l} = \frac{K/H}{(c_{p,w}\Delta T_{\mathrm{ev}} + L_v)} = \frac{4.1 \times 10^3}{2.67 \times 10^6} = 1.54 \times 10^{-3} \left[\frac{kg}{s\,K\,m^2}\right] \tag{A3}$$

$$K_{d,i} = \frac{K/H}{(c_{p,w}\Delta T_{\mathrm{ev}} + L_v + c_{p,i}\Delta T_m + L_f)} = \frac{4.1 \times 10^3}{3.03 \times 10^6} = 1.35 \times 10^{-3} \left[\frac{kg}{s\,K\,m^2}\right] \tag{A4}$$

where $c_{p,i} = 2.10 \times 10^3$ J K$^{-1}$ kg$^{-1}$ is the specific heat of ice at constant pressure and the latent heat of fusion $L_f = 3.34 \times 10^5$ J kg$^{-1}$. The equation for mass becomes $M = K_d \int_0^t A(t)(T_p - T_w(t))dt$.

Experiments comparing mass calculations using $\int_0^t A(t)T_p - T_w(t)dt$ and $\Delta T \int_0^t Adt$, where $\Delta T = T_p - T_w(t)$ is the mean value per particle showed that the latter is a sufficient approximation and the equation for mass may then be expressed as

$$M = K_d\Delta T \int_0^t A(t)dt \tag{A5}$$



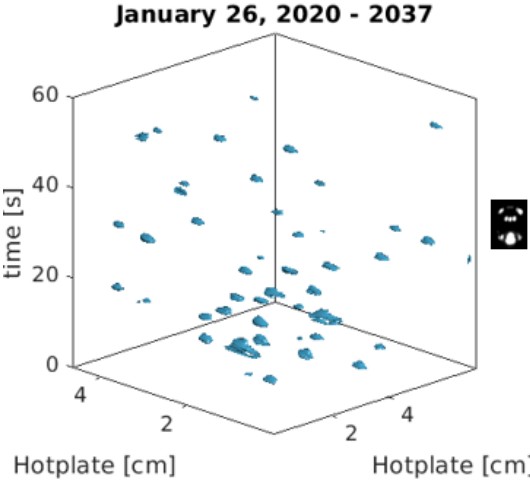

**Figure B1.** Volumetric rendering of melting rain hydrometeors in area on the DEID plate and in time such that each isosurface represents a rain drop frozen at a point in time.

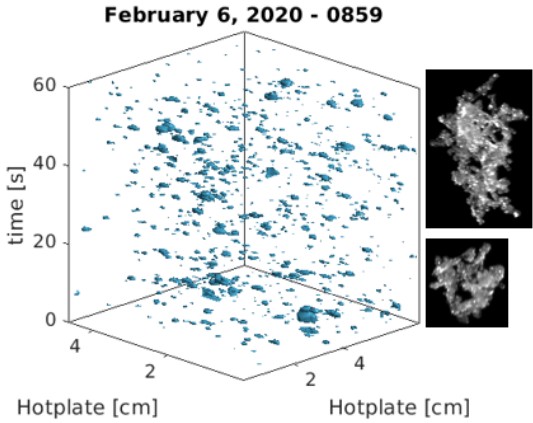

**Figure B2.** As for Fig. B1 but for partly melted hydrometeors

## Appendix B: Image processing

MATLAB Image Processing software was used to extract data from the thermal camera to determine the physical properties of individual melted hydrometeors. Experimentally relevant parameters include the hotplate temperature $T_p$, the sampling frequency of the thermal camera $f_s$, the physical width of each pixel in the camera imagery $p$, and the sampling area of the hotplate $A_{hot}$. Each thermal camera image was converted to both grayscale and binary format. 3-dimensional volumetric regions, "voxels," in a product space of hydrometeor area on the DEID plate and time during the duration of the melting of each particle were evaluated to yield the particle mass through Eq. A5, as illustrated in Figures B1-B5.





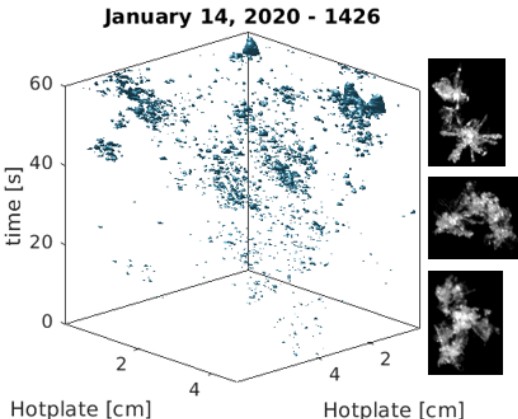

**Figure B3.** As for Fig. B1 but for aggregate snow

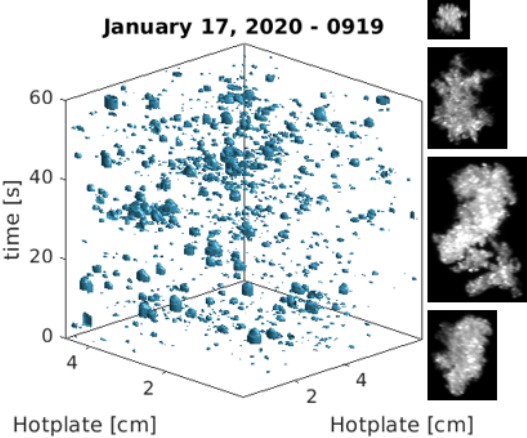

**Figure B4.** As for Fig. B1 but for densely rimed aggregate snow

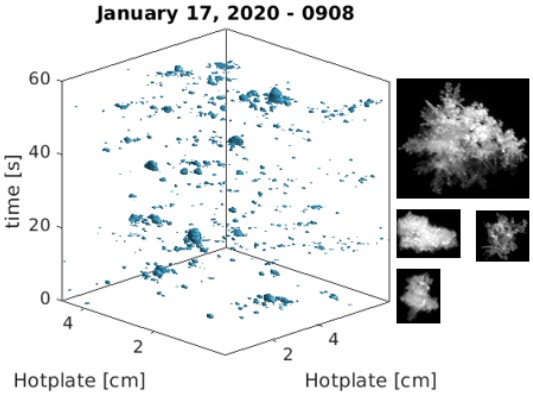

**Figure B5.** As for Fig. B1 but for a mixture of densely rimed and aggregate snow.





For the sake of processing, all partial voxel objects that bordered the edge of the 3D sampling area were removed for individual particle calculations but were included for precipitation rate calculations. A filtering threshold was used to remove small particles with fewer than three pixel data points in any dimension.

The temperature difference between the hotplate and water ($\Delta T = T_p - T_w(t)$) is based on the mean pixel intensity of the particle during its lifetime on the hotplate. The mean pixel intensity $I_{\mathrm{mean}}$ is converted to $\Delta T$ through the linear transformation ($(T_p - T_w(t)) \approx \Delta T = T_p(255 - I_{\mathrm{mean}})/256$ ).

The integrated cross-sectional area of the particle during its evaporation time on the hotplate is given by $V_t = \int A(t)dt$. The effective diameter $D_{\mathrm{eff}}$ is calculated from the point in time associated with the maximum recorded cross-sectional area of the

particle $A_{\mathrm{max}}$ according to $D_{\mathrm{eff}} = \sqrt{4A_{\mathrm{max}}/\pi}$. The evaporation time of each particle is calculated by generating a bounding box, or the smallest box that contains the 3D region containing $V_t$ where the circumscribed diameter $D_{\mathrm{max}}$ is the maximum in the area dimension and the evaporation time $t_{\mathrm{evap}}$ is the maximum in the time dimension.

**Appendix C: MASC snowflake imagery**

MASC Imagery of frozen hydrometeors was catalogued for each snow event described in this article with representative

particles shown in Figures C1 to C4



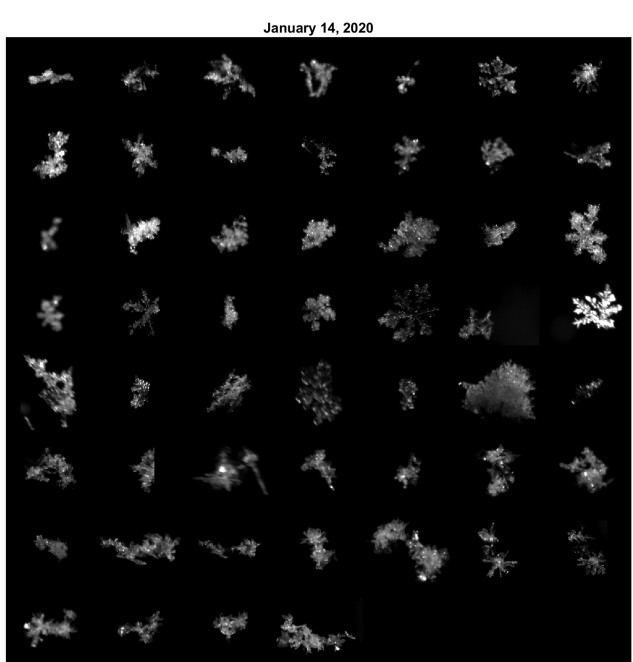

**Figure C1.** MASC Snowflake Imagery for January 14, 2020.





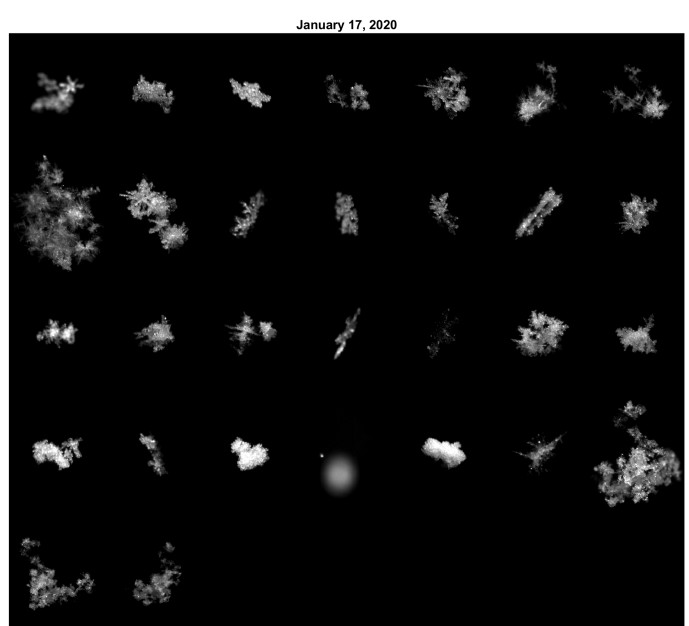

**Figure C2.** MASC Snowflake Imagery for January 17, 2020.



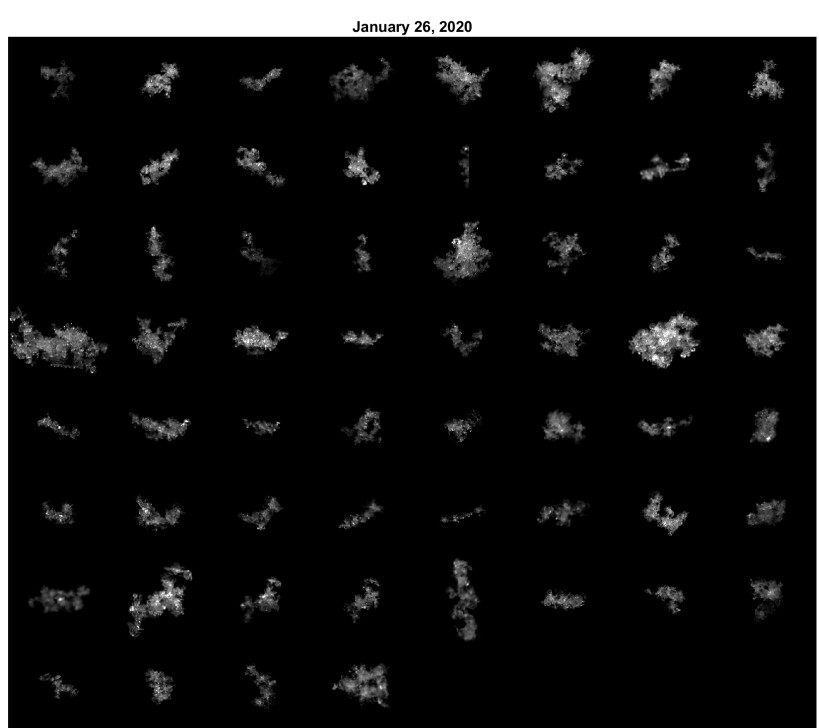

**Figure C3.** MASC Snowflake Imagery for January 26, 2020.



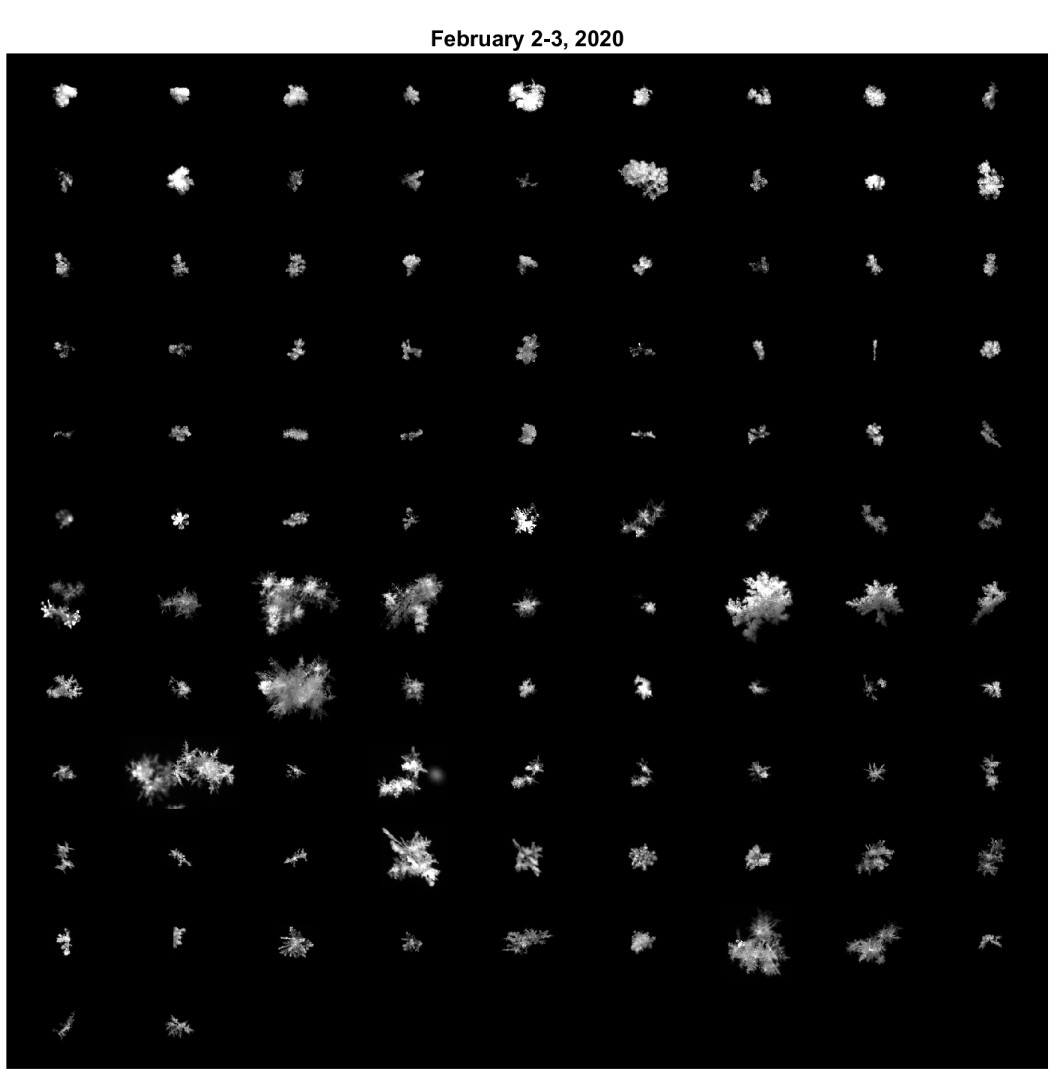

**Figure C4.** MASC Imagery for February 2 to February 3, 2020.



*Author contributions.* TJG and ERP conceived of the project. KNR and DS led collection and analysis of the data. KNR and TJG contributed equally to writing the manuscript with contributions from ERP.

*Competing interests.* The DEID is protected through a pending patent co-authored by KNR, DS, ERP, and TJG. TJG is a co-owner of Particle Flux Analytics, Inc. which has a licence from the University of Utah to commercialize the DEID.

*Acknowledgements.* This work is supported by the U.S. Department of Energy (DOE) Atmospheric System Research program award number DE-SC0016282 and National Science Foundation (NSF) Physical and Dynamic Meteorology program award number 1841870. We are grateful to Spencer Donovan, and Allan Reaburn and colleagues at Particle Flux Analytics for their contributions to the field program and the development of the DEID.





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
