# Peer review of "Measurement report: Mass and Density of Individual Frozen Hydrometeors"

_Atmospheric Chemistry and Physics, 2021_

## Referee Comment (RC2)

Title: Measurement report: Mass and Density of Individual Frozen Hydrometeors
Author(s): Karlie Rees et al.
MS No.: acp-2021-179
MS type: Measurement report

**General Comments:**

This paper analyzes measurements from a new instrument, the Differential Emissivity Imaging Disdrometer (DEID) to obtain continuous measurements of ice particle mass and effective size $D_{eff}$. The DEID data are combined with photographic imagery obtained using a Multi Angle Snowflake Camera (MASC) to obtain estimates of particle density. Results for three ice particle shapes are presented; graupel, densely rimed crystals and aggregates, where the number (N) of ice particles sampled in the latter two categories exceeds 15,000 (for each category). For graupel, N = 34. Mass-$D_{eff}$ and density-$D_{eff}$ power law relationships are presented for each shape category. The paper is well organized and well written with high-quality figures. It should be acceptable for publication after minor revisions. There are some concerns however that need to be addressed before publication, mentioned below and under Major Comments.

More information is needed for the ice particle shape categories of "densely rimed" and "aggregates". What is densely rimed; columnar crystals, planar crystals, or both? Does this include densely rimed aggregates? For aggregates, please indicate the type of primary component ice crystal, whether it is columnar or planar, and if columnar, whether it is short or long columns (or needles). This information may help explain why the power term $b$ is so large in the aggregate relationship $M = a\,D^b$ where $M$ is ice particle mass and $D = D_{eff}$.

Figure 12 in Chen and Lamb (1994, JAS) compares theoretical and observed values of the inherent ice crystal growth ratio $\Gamma^*$, from which $b$ can easily be calculated. Theory assumes prolate spheroids for columns and oblate spheroids for planar crystals, with the latter being relevant for hexagonal plates, broad branched dendrites and rimed planar crystals. Since there is reasonable agreement between theory and observations, their results provide constraints for likely values of $b$. For short and long columns/needles, $1.8 \leq b \leq 2.7$, while for the above noted planar crystals, $2.3 \leq b \leq 2.5$. While the DEID $b$ value for "densely rimed" conforms well with these ranges, the aggregates (DEID $b = 2.75$) would need to be comprised mostly of short columns to conform with the expected $b$ range, and short columns tend not to aggregate well. Thus, it is difficult to reconcile the $b$ value for aggregates with both theory and observations.

The authors compare their aggregate $b$ with aggregate $b$ values from Locatelli and Hobbs (1974), ranging from 1.4 (unrimed dendrites or side planes) to 1.9 (containing either side planes, columns & bullets or densely rimed dendrites). Given the component crystals, it makes sense that the latter value is larger (i.e., the increase in mass per unit size increase is larger). But it is hard to imagine packing the crystals so densely in an aggregate that $b = 2.75$.

Taking this a step further, Westbrook et al. (2004, "Theory of growth by differential sedimentation, with application to snowflake formation", Phys. Review E) presents a model of columnar particle aggregation based on the differential sedimentation of the particles. A condensation of these results are reported in Westbrook et al. (2004, "Universality in snowflake aggregation", GRL, **31**, L15104, doi:10.1029/2004GL020363, but the paper is difficult to understand due to missing information. They state that "The structure of the aggregates produced by this process is found to feed back on the dynamics in such a way as to stabilize both the exponents controlling the growth rate, and the fractal dimension of the clusters" (i.e., the value of $b$). Their model predicts $b = 2.05 \pm 0.1$, with theory giving $b = 2$. This is either close to or the same as the measured value of $b$ reported for all seven types of observed aggregates in Table 1 of Mitchell et al. (1990, "Mass-Dimensional Relationships for Ice Particles and the Influence of Riming on Snowfall Rates", JAM).

Overall, the evidence appears compelling for rejecting 2.75 as a plausible $b$ value for aggregates. Nonetheless, these are new and interesting measurements, and the community can decide how seriously to take them. But to make that decision, all of the above studies should be described and cited.

**Major Comments:**

1. Line 104: While this identity appears plausible, it is not convincing mathematically. Can this identity be demonstrated mathematically? Seems important since it is used to derive Eq. 3 below.

2. Table 1: Please add N (# samples) to this table.

3. Figure 7: It might be of interest that "heavily rimed dendrites" in Mitchell et al. (1990) have m = 0.068 $D^{2.2}$ in mg-mm units. This snow-type probably has more riming than the "densely rimed" category here (hence the larger prefactor), but $b$ is quite consistent with the Locatelli and Hobbs $b$ range. Erfani and Mitchell (2017, ACP) present evidence that riming changes the prefactor but not $b$.

4. Lines 189-191: This statement seems to contradict the findings of Chen and Lamb (1994, JAS) who show theoretically and observationally that the mass exponent for long columnar ice crystals is < 2 and lies between 2 and 2.5 for planar ice crystals.

5. Line 220: Please also provide the standard deviation values here.

6. Figure B3: Can this be understood as a 3-D volume showing the distribution of ice particles within that volume?  If so, can it be used to evaluate the PSD post-processing algorithms based on interarrival times, which are designed to reduce the contribution of shattered ice particles to the number concentration measured by optical probes?  The science question that might be addressed is whether "inertial clustering" of ice particles occurs naturally as it does for cloud droplets (Ray Shaw's work).  If ice particles tend to naturally cluster with relatively little space between particles, then interarrival algorithms may be "throwing the baby out with the bathwater" more often than is currently known.  While this is outside the scope of this paper, perhaps it might be worth looking into?

**Minor Comments:**

1. Line 74:  Eq: 2 => Eq. 2?

With best wishes for this paper,
David Mitchell

---

## Author Comment (AC1)

To the editor, Dr. Ervens:

We request that the manuscript be reclassified as a Research Article rather than a Measurement Report. As stated in the first sentence the significance of mass-diameter and density-diameter measurements of quite general significance:

"Predictions of precipitation amount, location, and duration have been shown to be especially sensitive to parameterized expressions for how fast a hydrometeor falls (Rutledge and Hobbs, 1984; Reisner et al., 1998; Hong et al., 2004; Fovell and Su, 2007; Lin et al., 2010; Liu et al., 2011; Iguchi et al., 2012; Theriault et al., 2012), affecting forecasts of hurricane trajectories (Fovell and Su, 2007) and storm lifetimes (Garvert et al., 2005; Colle et al., 2005; Milbrandt et al., 2010). From the perspective of fluid dynamics, fallspeed can be related to the mass and density of precipitation particles (Bohm, 1989)."

The measurements presented in this article describe the first direct automated measurements of hydrometeor mass-diameter and density-diameter relationships, including a total of 86,285 hydrometeors. For comparison, the widely used Locatelli and Hobbs (1974) parameterizations are based on 376 hydrometeors. The quantity is sufficient that we were able to characterize, as described in the revised manuscript, a dependence of the prefactor and exponent in the power-law relationships on riming and temperature.

Thus, the manuscript in its revised form presents measurements that we believe to have more general implications for the atmospheric sciences, not only for the results that were obtained, but due to the substantial methodological advance of the particle-by-particle hotplate technique.

Regards,

Karlie N. Rees
Dhiraj K. Singh
Eric R. Pardyjak
Timothy J. Garrett

Comment on acp-2021-179 Anonymous Referee #1 Referee comment on "Measurement report: Mass and Density of Individual Frozen Hydrometeors" by Karlie Rees et al., Atmos. Chem. Phys. Discuss., https://doi.org/10.5194/acp-2021-179-RC1, 2021

RC1: 'Comment on acp-2021-179', Anonymous Referee #1, 23 Apr 2021

We thank the reviewer for the constructive comments.

I support publication. The authors describe measurements of mass and density of thousands of frozen hydrometeors, during two months in 2020, using a newly developed instrument. A paper describing that instrument, the Differential Emissivity Imaging Disdrometer (DEID), is currently in review in AMTD (Singh et al.). While it isn't necessary to read both papers to understand this one, I did find a brief review of that paper helpful as I read through this one.

If the paper were essentially published as-is, it would be fine. I have a few comments that the authors can consider, but have no changes that I feel must be made for the manuscript to be publishable in ACP.

Lines 120-121: ``...raindrops do not preserve their area after impaction on the plate.''

line 89: "...frozen particles nonetheless tend to maintain their shape such that Amax is representative..."

A snowflake's impact on the hotplate is a function of two timescales: the contact time between the plate and the snowflake, and the melting time of the initial contacting layer of the snowflake. There is a competition between the contact time and melting time. Contact time decreases with increasing snowflake density, and melting time increases with increasing density. For a given snowflake density (74 kg m$^{-3}$), the contact time is O ($10^{-1}$ sec), and the melting time of a 100 μm thick layer is O ($10^{-3}$ sec).

When a snowflake melts, the normal reaction force of the surface to the snowflake is weakened. A roughened plate surface and the surface tension between the plate and initial melted water layer help to hold the snowflake in place after impacting the heated plate. Once it has melted completely, its shape on the plate is decided by gravity and surface tension. When surface tension dominates gravity, the shape of the snowflake ($A_{max}$) is nearly preserved before and after melting.

Lines 90-92 now read:

Nonetheless, due to surface tension, particles that are initially frozen tend to maintain their shape following melting so that $A_{max}$ is approximately representative of the frozen cross-sectional area in air. In calibration, Singh et al. (2021) found that snowflakes undergo only a 5% change in $D_{eff}$ during the melting process.

We observed for all snowflakes that the height of the water column is less than the capillary height after melting. Capillary height is defined as a maximum vertical height of water column accumulation on the hotplate before overflow (force balance between gravity and surface tension). The calculated capillary height of distilled water on a roughened aluminium plate is 1.987 mm, and in terms of the mass of water droplet/snowflake is approximately 100 mg.

line 211-212: "The plate was roughened with 600 grit sandpaper to allow for droplet spreading and more rapid evaporation."

Roughening the plate causes the contact angle of a water droplet to decrease and allows spreading. During water droplet experiments, roughening of the plate with 600 grit sandpaper allowed for droplet spread when the temperature of the plate was > 120° C or when a large mass (> 100 mg) was applied. Under operating conditions, however, the roughened surface did not contribute to a noticeable change in contact angle or area spread.

Lines 231-232 now read:

The plate was roughened with 600 grit (P1200) sandpaper to allow for droplet spreading and more rapid evaporation of water during calibration experiments.

These three statements introduce some ambiguity for me. I can see why droplet spreading and more rapid evaporation make the analysis of the heat transfer aspects of the problem easier, but why don't the droplets resulting from melting snowflakes or graupel spread? Why do they preserve their area (roughly)? Is it because they land on the plate rather gently? I know from my own observation of watching single snowflakes or aggregates land on surfaces that they don't tend to land forcefully. Usually the aggregates don't even break off pieces of the needles if they are clumps of dendrites. This makes sense to me because the terminal speed of these types of hydrometeors is usually pretty small.

Many raindrops, on the other hand, are large enough to have an appreciable terminal speed; the kinetic energy of their impact could flatten them. Does this mean that if drops are small enough (fog might be the limiting example) that their area would be preserved upon impact and the DEID could be used to estimate a density?

We agree with the reviewer's comments that larger drops spread more after impact, and the same trend was observed using the DEID. The estimated density of a ~1 mm diameter raindrop using mass and spherical volume is ~ 1000 kg m-3, which indicates that it is small enough to preserve its shape after impact.

There should be a relationship between the area that a drop makes upon contact and its size. Marshall and Palmer used a version of that phenomena when they correlated the size that drops made on dyed filter paper. (Before that, Bentley had used small canisters of flour to measure the sizes of raindrops.) It seems that something akin to that principle could be used here.

I recognize that this is beyond the scope of the present paper and that what I'm suggesting here is complicated by the fact that both liquid and frozen hydrometeors are impacting upon the plate. That said, perhaps the authors could comment on this, even if only in the Reply to the Reviews, since those are archived as well as the paper.

Citation: https://doi.org/10.5194/acp-2021-179-RC1

We thank the reviewer for the constructive comments.

General Comments:

This paper analyzes measurements from a new instrument, the Differential Emissivity Imaging Disdrometer (DEID) to obtain continuous measurements of ice particle mass and effective size $D_{eff}$. The DEID data are combined with photographic imagery obtained using a Multi Angle Snowflake Camera (MASC) to obtain estimates of particle density. Results for three ice particle shapes are presented; graupel, densely rimed crystals and aggregates, where the number (N) of ice particles sampled in the latter two categories exceeds 15,000 (for each category). For graupel, N = 34. Mass-$D_{eff}$ and density-$D_{eff}$ power law relationships are presented for each shape category. The paper is well organized and well written with high-quality figures. It should be acceptable for publication after minor revisions. There are some concerns however that need to be addressed before publication, mentioned below and under Major Comments.

More information is needed for the ice particle shape categories of "densely rimed" and "aggregates". What is densely rimed; columnar crystals, planar crystals, or both? Does this include densely rimed aggregates?

Densely rimed aggregates were initially classified as aggregates. Upon review, we found that the January 26, 2020 storm comprised 11,080 large, aggregate snowflakes, which were initially classified as aggregates but perhaps are better classified as "densely rimed." The densely rimed category contains all snowflakes not categorized as graupel or aggregates, similar to Garrett and Yuter (2014) who used the MASC derived complexity parameter, $\chi = P(1 +< \sigma >)/(2\pi r)$ where P is snowflake perimeter, r is snowflake radius, and $< \sigma >$ is intensity variability, to classify snowflakes into three categories: graupel ($\chi<1.35$), densely rimed ($1.35\leq\chi\leq1.75$), and aggregates ($\chi>1.75$).

For aggregates, please indicate the type of primary component ice crystal, whether it is columnar or planar, and if columnar, whether it is short or long columns (or needles). This information may help explain why the power term b is so large in the aggregate relationship M = a $D_b$ where M is ice particle mass and D = $D_{eff}$.

Added to lines 158-160:

The densely rimed category includes all snowflakes not categorized as graupel or aggregates following Garrett and Yuter (2014). It also includes densely rimed aggregates and partially melted aggregates. Figures 3 and C1-C4 show mostly planar type crystals present, but aggregated needles are frequently seen as well.

Figure 12 in Chen and Lamb (1994, JAS) compares theoretical and observed values of the inherent ice crystal growth ratio Γ*, from which b can easily be calculated. Theory assumes prolate spheroids for columns and oblate spheroids for planar crystals, with the latter being relevant for hexagonal plates, broad branched dendrites and rimed planar crystals. Since there is reasonable agreement between theory and observations, their results provide constraints for likely values of b. For short and long columns/needles, $1.8 \leq b \leq 2.7$, while for the above noted planar crystals, $2.3 \leq b \leq 2.5$. While the DEID b value for "densely rimed" conforms well with these ranges, the aggregates (DEID b = 2.75) would need to be comprised mostly of short columns to conform with the expected b range, and short columns tend not to aggregate well. Thus, it is difficult to reconcile the b value for aggregates with both theory and observations.

The authors compare their aggregate b with aggregate b values from Locatelli and Hobbs (1974), ranging from 1.4 (unrimed dendrites or side planes) to 1.9 (containing either side planes, columns & bullets or densely rimed dendrites). Given the component crystals, it makes sense that the latter value is larger (i.e., the increase in mass per unit size increase is larger). But it is hard to imagine packing the crystals so densely in an aggregate that b = 2.75.

The large fraction of snowflakes from the January 26, 2020 storm were largely partially melted (Figure C3), which likely contributed to such a large value of b=2.75. The mass-diameter relationship was recalculated (below) for snowflakes binned by MASC-derived snowflake complexity. The blue curves represent the same analysis where partially melted snowflakes were excluded using the ambient air temperature.

[Figure]

While the value of b does drop to 2.2 for the most complex and least rimed snowflakes, its relationship is not clearly dependent on complexity in this analysis. Although, filtering out snowflakes that occurred when the ambient air temperature was above -3C resulted in b values substantially lower than previously obtained. The value of b is between 2.0-2.3 for all complexities when the air temperature is <-3C, and ranges from 2.2-2.7 for all air temperatures in this dataset, which are up to 1.5C. Both temperature curves are shown in this response and for clarity, Figure 9 was added to the manuscript, which has just the T<-3°C case.

The values of a and b in the **mass-diameter** relationship for all snowflakes are as follows:

T > 0C: a=0.02 b=2.80
T < 0C: a=0.02 b=2.33
T <-2C: a=0.02 b=2.38
T <-3C: a=0.01 b=2.12

Added Table 3 and the following to lines 197-198:

The values of a and b were also obtained when filtering snowflakes by temperature to exclude partially melted snowflakes (T<0°C) and to reflect primarily aggregate snowflakes (T<-3°C) and are shown in Table 3.

Taking this a step further, Westbrook et al. (2004, "Theory of growth by differential sedimentation, with application to snowflake formation", Phys. Review E) presents a model of columnar particle aggregation based on the differential sedimentation of the particles.  A condensation of these results are reported in Westbrook et al. (2004, "Universality in snowflake aggregation", GRL, 31, L15104, doi:10.1029/2004GL020363, but the paper is difficult to understand due to missing information.  They state that "The structure of the aggregates produced by this process is found to feed back on the dynamics in such a way as to stabilize both the exponents controlling the growth rate, and the fractal dimension of the clusters" (i.e., the value of b).  Their model predicts b = 2.05 ± 0.1, with theory giving b = 2.  This is either close to or the same as the measured value of b reported for all seven types of observed aggregates in Table 1 of Mitchell et al. (1990, "Mass-Dimensional Relationships for Ice Particles and the Influence of Riming on Snowfall Rates", JAM).

Overall, the evidence appears compelling for rejecting 2.75 as a plausible b value for aggregates.  Nonetheless, these are new and interesting measurements, and the community can decide how seriously to take them.  But to make that decision, all of the above studies should be described and cited.

 We thank the reviewer for pointing out the b value of 2.75, which appears to be the result of a large fraction of partially melted aggregate snowflakes included in the aggregate category. We have recategorized those flakes into the densely rimed category and included a figure with mass-diameter relationships and a table with density-diameter relationships as filtered by ambient temperature.

Added to lines 170-186:

Figure 9 illustrates the sensitivity of mass-diameter relationship parameters to particle type and ambient air temperature. Garrett and Yuter (2014) employed the MASC-derived complexity parameter, $\chi = P(1 +< \sigma >)/(2\pi r)$ where P is snowflake perimeter, r is snowflake radius, and $< \sigma >$ is intensity variability, to classify snowflakes into three categories: graupel ($\chi<1.35$), densely rimed ($1.35\leq\chi\leq1.75$), and aggregates ($\chi>1.75$). Threshold values of 1.3 and 1.8 have also been previously used (Garrett et al. (2015). Here we adopt threshold values of 1.3 and 1.7. Graupel-like snow is included in the graupel category; the two datapoints with $\chi<1.3$ include 9,506 snowflakes, which are unlikely to be solely graupel due to one-minute averaging of $\chi$ and the presence of other snow types observed during graupel events. This category has values of $b$ consistent with the exponent values of between 2.1 and 2.4 found by Locatelli and Hobbs (1974). The densely rimed category however has exponent values between 2.5 and 2.7, higher than those seen by Locatelli and Hobbs (1974), although the difference may be influenced by presence of a large number of partially melted snowflakes that bring the exponent closer to 3. Overall, smaller values of $b$ are obtained as $\chi$ exceeds 1.7 and snowflakes transition into aggregates. Notably, the value of $b$ is never lower than 2.

Partially melted snowflakes are excluded favoring more aggregate-type snowflakes by restricting analysis to particles that fell when the ambient air temperature was <-3°C, as represented by blue lines in Figure 9. There is a clear sensitivity in the mass-diameter relationships to ambient air temperature. For all snowflakes that occurred when the ambient air temperature was < 0°C (N=30,651), the values of $a$ and $b$ are 0.017 and 2.33 respectively with $R^2$=0.85. For all snowflakes that fell when the ambient air temperature was < -3°C (N=4,630), the corresponding values are 0.015 and 2.12 with $R^2$=0.84.

Major Comments:

1. Line 104: While this identity appears plausible, it is not convincing mathematically. Can this identity be demonstrated mathematically? Seems important since it is used to derive Eq. 3 below.

   Added Figure 4 to clarify diameter and volume measurements and calculations

1. Table 1: Please add N (# samples) to this table.

   Added N to Table 1.

1. Figure 7: It might be of interest that "heavily rimed dendrites" in Mitchell et al. (1990) have m = 0.068 $D_2$ in mg-mm units. This snow-type probably has more riming than the "densely rimed" category here (hence the larger prefactor), but b is quite consistent with

the Locatelli and Hobbs b range. Erfani and Mitchell (2017, ACP) present evidence that riming changes the prefactor but not b.

Added to lines 167-169: In contrast to the findings of Erfani and Mitchell (2017) which state that particle riming changes the prefactor *a* but not *b*, here both *a* and *b* decrease with increased riming.

1. Lines 189-191: This statement seems to contradict the findings of Chen and Lamb (1994, JAS) who show theoretically and observationally that the mass exponent for long columnar ice crystals is < 2 and lies between 2 and 2.5 for planar ice crystals.

    Removed this statement to be consistent with updated results.

1. Line 220: Please also provide the standard deviation values here.

    Added standard deviation values to line 241: 4.1±0.01603

1. Figure B3: Can this be understood as a 3-D volume showing the distribution of ice particles within that volume? If so, can it be used to evaluate the PSD post-processing algorithms based on interarrival times, which are designed to reduce the contribution of shattered ice particles to the number concentration measured by optical probes? The science question that might be addressed is whether "inertial clustering" of ice particles occurs naturally as it does for cloud droplets (Ray Shaw's work). If ice particles tend to naturally cluster with relatively little space between particles, then interarrival algorithms may be "throwing the baby out with the bathwater" more often than is currently known. While this is outside the scope of this paper, perhaps it might be worth looking into?

    The 3D volumes do reveal both spatial and temporal clustering. Non-poissonian droplet clustering is detectable by the DEID, and is briefly addressed in our paper, *Idealized simulation study of the relationship of disdrometer sampling statistics to the precision of precipitation rate measurement*. Further studies of hydrometeor clustering is a novel application of the DEID.

Minor Comments:

1. Line 74: Eq: 2 => Eq. 2?

    Line 74 now reads: Eq. 2

With best wishes for this paper,

David Mitchell

Citation: https://doi.org/10.5194/acp-2021-179-RC2

References

Garrett, T. J., and Yuter, S. E. (2014), Observed influence of riming, temperature, and turbulence on the fallspeed of solid precipitation, *Geophys. Res. Lett.*, 41, 6515– 6522, doi:10.1002/2014GL061016.

Garrett, T. J., Yuter, S. E., Fallgatter, C., Shkurko, K., Rhodes, S. R., and Endries, J. L. (2015), Orientations and aspect ratios of falling snow. *Geophys. Res. Lett.*, 42, 4617– 4622. doi: 10.1002/2015GL064040.

Rees, K. and Garrett, T. J.: Effect of disdrometer sampling area and time on the precision of precipitation rate measurement, Atmos. Meas. Tech. Discuss. [preprint], https://doi.org/10.5194/amt-2020-393, in review, 2020.